# NAIL: Lexical Retrieval Indices with Efficient Non-Autoregressive Decoders

**Livio Baldini Soares**     **Daniel Gillick**     **Jeremy R. Cole**     **Tom Kwiatkowski**

Google Deepmind

{liviobs,jrcole,dgillick,tomkwiat}@google.com

## Abstract

Neural document rerankers are extremely effective in terms of accuracy. However, the best models require dedicated hardware for serving, which is costly and often not feasible. To avoid this serving-time requirement, we present a method of capturing up to $86\%$ of the gains of a Transformer cross-attention model with a lexicalized scoring function that only requires $10^{-6}\%$ of the Transformer's FLOPs per document and can be served using commodity CPUs. When combined with a BM25 retriever, this approach matches the quality of a state-of-the art dual encoder retriever, that still requires an accelerator for query encoding. We introduce NAIL (**N**on-**A**utoregressive **I**ndexing with **L**anguage models) as a model architecture that is compatible with recent encoder-decoder and decoder-only large language models, such as T5, GPT-3 and PaLM. This model architecture can leverage existing pre-trained checkpoints and can be fine-tuned for efficiently constructing document representations that do not require neural processing of queries.

## 1 Introduction

We attempt to answer the following question: to what extent can the computationally-intensive inference in modern neural retrieval systems be pushed entirely to indexing time?

Neural networks have revolutionized information retrieval, both with powerful reranking models that cross-attend to query and document, and with dual-encoder models that map queries and documents to a shared vector space, leveraging approximate nearest neighbor search for top-k retrieval. The strongest systems typically use a dual-encoder for retrieval followed by a cross-attention reranker to improve the ordering. However, both these components tend to be built on increasingly large Transformers (Ni et al., 2021; Nogueira dos Santos et al., 2020; Izacard et al., 2021; Hui et al., 2022) and thus rely on dedicated accelerators to process queries

quickly at serving time. In many application settings, this may be impractical or costly, and as we will show, potentially unnecessary.

In particular, we explore a retrieval paradigm where documents are indexed by predicted query token scores. As a result, scoring a query-document pair $(q, d)$ simply involves looking up the scores for the tokens in $q$ associated with $d$ in the index. While the scores are predicted by a neural network, the lookup itself involves no neural network inference so can be faster than other approaches. However, this also means that there can be no cross-attention between a specific query and document or even a globally learned semantic vector space. Given these shortcomings, it is unclear that such a model, which offloads all neural network computation to indexing time, can be a practical alternative to its more expensive neural counterparts.

In addition, while large pre-trained language models have been shown to generalize well over a number of language and retrieval tasks (Chowdhery et al., 2022; Raffel et al., 2020; Brown et al., 2020; Nogueira et al., 2019b; Ni et al., 2021), a key challenge is that they have universally adopted a sequence-to-sequence architecture which is not obviously compatible with precomputing query scores. Naive approaches are either computationally infeasible (scoring all possible queries), or rely on sampling a small, incomplete set of samples (such as in Lewis et al. 2021).

To overcome this challenge, we introduce a novel use of non-autoregressive decoder architecture that is compatible with existing Transfomer-based language models (whether Encoder-Decoder or Decoder-only, Chowdhery et al. 2022). It allows the model, in a single decode step, to score all vocabulary items in parallel. This makes document indexing with our model approximately as expensive as indexing with document encoders used in recent dual-encoder retrieval systems (Ni et al., 2021; Izacard et al., 2021; Formal et al., 2021a). We call the retrieval system based on this proposed

| System | Cross-Attention Enc. | | | Query/Dual Enc. | | | | Lexical | |
|---|---|---|---|---|---|---|---|---|---|
| | MonoT5-3B | MiniLM-L6 | TinyBERT-L6 | GTR-XXL | Contriever | Splade-v2 | BERT-tiny | Splade-doc | NAIL |
| | $10^{11}$ | $10^{10}$ | $10^{10}$ | $10^{11}$ | $10^9$ | $10^9$ | $10^8$ | $10^2$ | $10^2$ |

Table 1: Estimated FLOPS required to score a $(query, document)$ pair, using estimators by Clark et al. (2020). For dual-encoder and lexical systems, document representations are precomputed. $query$ is assumed to be of length 16 tokens, and $document$ is assumed length of 128 tokens. The standard versions of Splade-v2 and Contriever are based on BERT-base.

model NAIL (**N**on-**A**utoregressive **I**ndexing with **L**anguage models). We summarize our contributions as follows:

1. We advance prior work on learned sparse retrieval by leveraging pretrained LMs with a novel non-autoregressive decoder.

2. We describe a range of experiments using the BEIR benchmark (Thakur et al., 2021) that explore the performance and efficiency of our model as a reranker and as a retriever. As a reranker, NAIL can recover 86% of the performance of a large cross-attention reranker (Nogueira et al., 2020), while requiring $10^{-6}$% of the inference-time FLOPS. As a retriever, NAIL has an extremely high upper bound for recall—exceeding the performance of all other retrievers in the zero-shot setting. Finally, by using BM25 as a retriever and NAIL as a reranker, we can match state-of-the-art dual-encoders (Ni et al., 2021; Izacard et al., 2021) with $10^{-4}$% of the inference-time FLOPS.

3. We propose our model as a preferred solution when significant compute is available at indexing time, but not on-demand at serving time, and we provide a cost analysis that illustrates when our approach could be preferred to previous work that harnesses LLMs.

## 2 Related work

There has been much work in information retrieval leveraging neural networks, which we cannot adequately cover in this paper. For a comprehensive overview, we refer the reader to the survey by Hambarde and Proenca 2023. Here, we focus on methods that minimize the use of expensive neural methods at query inference time (typically methods of *sparse retrieval*) and on those that leverage LLMs.

**LM-based Term Weighting**   Bag-of-words models, such as TF-IDF and BM25 (Robertson and Zaragoza, 2009), use term weighting based on corpus statistics to determine relevance of document terms to query terms. Our work can be seen as a way to construct document term weights that are both (1) unconditional with respect to the query, and (2) indexed using lexicalized features (specifically, we use a vector of token scores). As a result, this type of document representation can be precomputed (at indexing time) and does not require expensive computation at query-time. Prior work on leveraging language models to produce such lexicalized term weighting can be roughly divided into two groups: those with just document-side encoders, and those with query-side and document-side encoders.

Examples of the first group include DeepCT (Dai and Callan, 2020), DeepTR (Zheng and Callan, 2015), and DeepImpact (Mallia et al., 2021), Tilde v2 (Zhuang and Zuccon, 2021), and Splade-doc (Formal et al., 2021a). These systems are examples of the model paradigm we are exploring, in which all neural network computation happens at indexing time. Our work can be seen as an attempt to update these systems (which use word2vec embeddings or encoder-only language models) to modern encoder-decoder architectures. Splade-doc is the most recent (and performant) of these, so is in many cases the most useful point of comparison for our work. We include results for the best version of Splade-doc (Lassance and Clinchant, 2022).

Examples of the second group include SPARTA (Zhao et al., 2021), ColBERT (Khattab and Zaharia, 2020), ColBERT v2 (Santhanam et al., 2022), COIL (Gao et al., 2021), Splade (Formal et al., 2021b), and Splade v2 (Formal et al., 2021a). These sparse dual-encoders have proven themselves competitive with dense dual-encoders, and have some advantages like improved interpretability. We demonstrate comparable performance without the need for any query-side encoder.

**LM-based Document Expansion**   Another way to improve retrieval indices using language models is document expansion. This consists of augmenting the terms in a document that do not occur in its original text, but are likely to be useful for retrieval. When used in combination with a lexicalized retrieval index, document expansion can be implemented without additional query-

time computational requirements. Recent examples of LM-based document expansion systems include Doc2Query (Nogueira et al., 2019c) and Doc2Query-T5 (Nogueira et al., 2019a).

Other forms of document expansion include the *Probably asked questions* database (Lewis et al., 2021) which, via an expensive offline system, uses a generative language model to produce lists of questions for every document in the corpus.

We agree with Lin and Ma (2021) that document expansion typically improves the quality of retrieval systems, irrespective of representation used. Our approach, however, makes no assumptions about which terms should be used to index a document, allowing the model to score all tokens in the vocabulary.

**Non-autoregressive decoders**   Non-autoregressive sequence-to-sequence models have been previously proposed and studied, particularly in the context of machine translation (Gu et al., 2018; van den Oord et al., 2018; Lee et al., 2018), motivated by the computational complexity of standard auto-regressive decoding, which requires a decode step per generated token. Non-autoregressive decoding breaks the inter-step dependency and thus provides two computational benefits: (1) a single step through the decoder can produce outputs for more than one position, and (2) computation can be easily parallelized since are is no time-wise dependencies between computations.

While these systems use non-autoregressive decoding to perform iterative generation of text, we know of no existing work that uses non-autoregressive decoding to produce document representations or for retrieval purposes.

## 3   NAIL **Model**

A major goal of this work is to investigate retrieval methods that forego neural computation and the need for specialized accelerator hardware *at query time*. As such, we focus on a method that uses a large neural model to precompute the required representations of the retrieval items (documents) ahead of time. Then, at retrieval time, the method performs only basic featurization (e.g., tokenization) of the queries.

Specifically, we investigate query-document scoring functions that score the compatibility of a query-document pair with the inner-product of separate featurizations of the query $\phi_q(q)$ and document $\phi_d(d)$.

$$\text{score}(q, d) = \langle \phi_q(q), \phi_d(d) \rangle \qquad (1)$$

This form is familiar from both traditional lexicalized retrieval and from more recent work on dense retrieval. In lexicalized retrieval, (e.g., TF-IDF and BM25) (Robertson and Zaragoza, 2009; Robertson and Walker, 1994), $\phi_q$ and $\phi_d$ assign non-zero scores to sub-strings of $q$ and $d$. On the other hand, in dense retrieval (Karpukhin et al., 2020; Ni et al., 2021; Izacard et al., 2021), $\phi_q$ and $\phi_d$ are neural networks that map $q$ and $d$ to dense vectors. Note that this formulation does not allow for deeper interactions between $d$ and $q$, such as cross-encoder scorers, as these cannot be computed efficiently and without an accelerator at query time.

We investigate an alternative formulation of Equation 1 than either traditional lexicalized retrieval or dense retrieval. In this formulation, $\phi_d$ can be an arbitrarily complex neural network, but $\phi_q$ must be a sparse featurization that can be quickly computed on commodity CPUs. This way, it is possible to push all costly neural network inference to indexing time, and avoid the need for accelerators at serving-time. For this paper, we choose $\phi_q$ to be a simple tokenizer, but we believe that our results could also extend to more complex sparse featurizations.

### 3.1   Independent prediction of query tokens

Given the choice of $\phi_q$ described above, we need to learn a function $\phi_d$ that can assign high scores to tokens that are are likely to occur in a query associated with the input document and low scores to tokens that are unlikely to appear in such a query. This goal differs from related work on query prediction for document expansion (Nogueira et al., 2019b; Lewis et al., 2021) where only a few likely query terms are added to the set of document terms.

Instead of aiming to predict a small number of queries that are related to $d$, we aim to predict a featurization of $d$ that can be used to score *any* query. Given that an important motivation of this work is to make use of large pretrained language models, we must also investigate how best to adapt the sequence-to-sequence generative architecture that most such models have adopted. In particular, the Transformer-based language models adopt an autoregressive decoding strategy, where the model predicts a single token position at a time, conditioned on the output of previous predictions. A naive decoding strategy, of decoding every possible target query ahead of time, is not computationally feasible, requiring $32k^{16} = 10^{72}$ decode steps (or more generally, $|\mathcal{V}|^l$, where $\mathcal{V}$ is the vocabulary and $l$ is the length of the query).

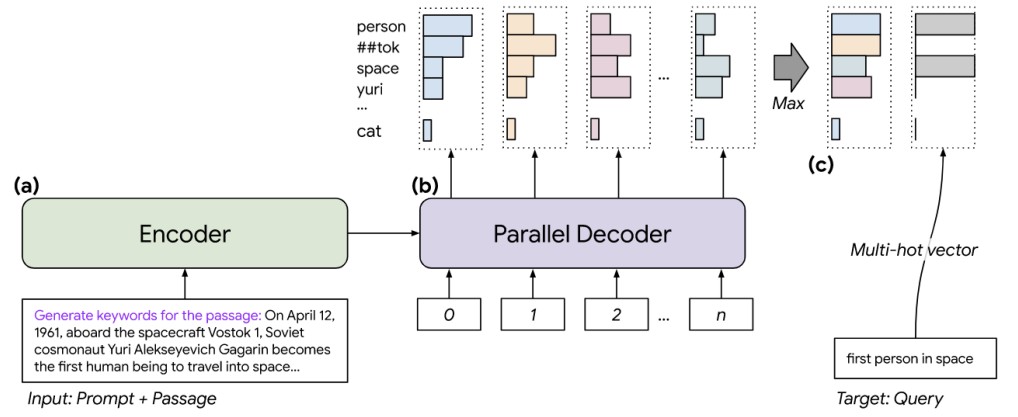

Figure 1: Our model adapts the T5 encoder-decoder architecture to predict query token scores given an input passage. The encoder *(a)* reads an input passage. The decoder *(b)* is initialized from a pretrained T5 checkpoint, but the architecture is modified in a few ways to be non-autoregressive: the only inputs are the standard position embeddings, the decoding is parallelized for efficiency, and the output at each position is the full distribution over the vocabulary. Finally, we take a max over the position axis *(c)* to produce a vector of token scores corresponding to the multi-hot vector of tokens appearing in the target query.

*How do we generate document representations, using a sequence-to-sequence architecture, in a computationally efficient way?*

To do this, while also making use of pre-trained Transformer language models, we modify the decoder stack to support independent predictions of the output tokens (also known in the literature as *non-autoregressive decoding*, Lee et al. 2018; Gu et al. 2018). In addition, we modify the output of the model so that instead of generating a token sequence, it generates a sequence of scores over the vocabulary. We use this predicted sequence of vector of scores over the vocabulary as a representation of the document $d$ in our system.

Our model architecture is illustrated in Figure 1. In this model, each output token is predicted independently from other output tokens, and is conditioned only on input sequence and positional information. This allows the model to produce output for all positions in parallel. In addition, because the output representation is no longer a single token, but scores over the entire vocabulary, we can obtain a representation for scoring any possible query $q$ in a single step of the decoder.

The NAIL model is based on the T5 architecture (Raffel et al., 2020) and, for the experiments in Section 5, we start with pre-trained T5 checkpoints. There are several ways to use such a model to predict feature scores. NAIL uses the T5 vocabulary as its featurization, consisting of 32,000 tokens. In order to quickly score all 32,000 tokens, we modify the baseline model in two ways:

1. The standard encoder-decoder model proceeds auto-regressively, predicting the next token based on the previous predicted tokens. Each output token additionally conditions on a relative position embedding based on the current decode position. Here, instead there are a fixed number of decode positions which all proceed simultaneously, conditioning only on the input and a fixed position embedding.

2. In both the standard T5 model and our adaptation of it, each token position outputs a distribution over the entire output vocabulary. Normally, this produces a single sequence of tokens by sampling or taking the maximum probability token at each position. Here, we instead pool over all positions, taking the maximum token score produced at any position.

A simpler alternative would be to have the model decode for only a single position and then use the produced distribution as the scores for each token. However, we found that the model was able to represent a more diverse and better-performing distribution of query tokens when it could distribute their predictions over multiple output positions.

## 3.2 Contrastive training

Similar to previous work that has trained dual encoders for retrieval, we utilize negative training examples in order to do contrastive learning. In particular, we assume training data of the form $\mathcal{D} = \{(q_0, d_0^+, \mathbf{d}_0^-), \ldots, (q_n, d_n^+, \mathbf{d}_n^-)\}$ made up of triples that associate a query $q_i$ with a positive passage $d_i^+$ and a set of $k$ negative passages $\mathbf{d}_i^- = \{d_{i:0}^-, \ldots, d_{i:k}^-\}$. The negative passages are typically related to the query but are worse retrievals than the positive passages.

We train NAIL by assembling $\mathcal{D}$ into batches of $m$ examples and calculating an in-batch softmax that includes both positive and negative passages from the batch (Ni et al., 2021). Let a single batch of $m$ examples be

$$\mathbf{b}_i = ((q_{i*m}, d^+_{i*m}, d^-_{i*m}), \ldots,$$
$$(q_{i*m+m-1}, d^+_{i*m+m-1}, \mathbf{d}^-_{i*m+m-1})) \qquad (2)$$

and let $\mathbf{d}_i$ be all of the positive and negative candidate passages in this batch. The per-example loss for a query $q$ and positive passage $d^+$ drawn from batch $\mathbf{b}_i$ is

$$\mathcal{L} = -\langle \phi_q(q_i), \phi_d(d^+) \rangle + \log \sum_{d' \in \mathbf{d}_i} \exp(\langle \phi_q(q_i), \phi_d(d') \rangle) \qquad (3)$$

and we train the model to incrementally minimize the per-batch loss, summed over all $m$ examples in the batch. Note that the number of explicit negative passages can vary under this setup, as the positive passages for other queries serve as implicit negative passages for every other query. More details about the training setup are given in the following section.

## 4 Model Training and Experiments

To train the NAIL model, we have empirically found it beneficial to perform two stages of training (1) a pre-training stage the uses self-supervised tasks over a large, unlabeled text corpus, and (2) a fine-tuning stage that relies on question-answering data via explicit hard negatives. We present the details of each of the training steps in Sections 4.1 and 4.2.

Our model is implemented within the T5X framework (Roberts et al., 2022) and we initialize model weights with published T5.1.1 checkpoints (Raffel et al., 2020). Unless otherwise noted, the NAIL model size used in the experiments is *XL*, with roughly 3 billion parameters. We saw no further gains from increasing parameters further.

To be compatible with T5 checkpoints, we also adopt the T5 vocabulary and attendant SentencePiece tokenizer (Kudo and Richardson, 2018). The vocabulary consists of 32,000 tokens extracted from a English-focused split of Common Crawl.

### 4.1 Pre-training

For pretraining, we combine two related self-supervision tasks for retrieval: inverse cloze and independent cropping (Lee et al., 2019; Izacard et al., 2021). Both of these tasks take in a passage from a document and generate a pair of spans of text, forming a positive example. One of the generated spans serves as a pseudo-query and the other as a pseudo-passage. In independent cropping, two contiguous spans of text are sampled from the passage. As the spans are selected independently, overlaps between them are possible. For the inverse cloze task, a contiguous span is initially selected from the passage, forming a pseudo-query. The second span encompasses the remainder of the passage with the sub-sequence selected in the first span omitted.

In both tasks, we use the C4 corpus (Raffel et al., 2020), a cleaned version of Common Crawl's web crawl corpus. In each training batch, half of the examples are from the independent cropping task and half are from the inverse cloze task. In addition, each target has a single correct corresponding input, and all other inputs serve as negatives.

We found this pre-training to be very important to calibrate language model scores to lexical retrieval scores. One possible reason is that while highly frequent words (*stop words*) typically have a high score in LMs, they are known to be insignificant or harmful in ranking retrievals independent of the context or inputs in which they occur. Additional discussion of the need for pre-training can be found in Appendix B.2. We run pre-training for 500k steps on batches of 2048 items, the largest size we are able to fit into accelerator memory.

### 4.2 Fine-tuning

We finetune our model on the MS-MARCO dataset (Nguyen et al., 2016). It consists of roughly 500,000 queries, each with a corresponding set of gold passages (typically one per query) along with a set of 1,000 negative passages produced by running a BM25 system over the full corpus of 8.8M passages. We construct training examples using the gold passage as positive, along with a sample of the BM25 candidate passages as hard negatives.

We investigate a variable number of MS-MARCO hard negatives and find that more hard negatives improves MS-MARCO performance but worsens BEIR performance. More details can be found in Appendix B.1. Similar to pre-training, each batch consists of 2048 total passages.

### 4.3 Evaluation Methodology

For evaluation, we focus on the public, readily-available, datasets available in the BEIR (Thakur et al., 2021) suite and which have baseline numbers present in the leaderboard, which totals 12 distinct datasets. We specifically target BEIR since it contains a heterogeneous set of retrieval datasets, and equally importantly, evaluates these datasets in zero-shot setting. While neural models have made

huge gains over BM25 on *in-domain* data, BEIR shows that a variety of neural retrievers underperform relative to BM25 on *out-of-domain* data.

BEIR results are typically presented as two separate tasks, where most systems are only evaluated on either the *reranking* variant or the *full retrieval* variant. In the full retrieval variant, systems must retrieve over the provided corpus of document passages, which range from a few thousand to a few million, and they are evaluated on their recall@100 and their nDCG@10 (Järvelin and Kekäläinen, 2002), providing a view into their ability to retrieve the gold passages into the top 100 and the ordering of the top ten passages, respectively. In the reranking variant, models do not have to do retrieval, and the recall@100 is fixed to the performance of an off-the-shelf BM25 system, so only nDCG@10 is reported.

## 5 Experimental Evaluation

We compare NAIL to other systems that have published results on BEIR. To compare with some sparse systems that have not been evaluated on BEIR datasets, we also make of use the MS-MARCO passage ranking task. We focus on answering the following questions:

- How does NAIL perfom as a reranker, particularly when compared to much more expensive neural reranker systems?
- How does NAIL compare to recent term weighting retrieval systems that use language models?
- How does NAIL compare with a similarly trained dual-encoder system that uses an expensive query-side encoder?

Further experimental work is also presented in appendices, including: qualitative analysis (Appendix A), sensitivity to hard-negatives in the batch loss (Appendix B.1), effects of ablating pre-training or fine-tuning (Appendix B.2), and analysis of sparsifying document representations to make them more efficient for indexing (Appendix C).

### 5.1 Reranking

In the *reranking* BEIR task, each system must rerank the 100 passages returned by an off-the-shelve BM25 system.

**Baselines** In this section we divide approaches into two types of systems: lexical-based approaches and cross-encoders. In the cross-encoder category, we compare to MonoT5-3B (Nogueira

| nDCG@10 | Cross Enc. | | Lexical | |
|---|---|---|---|---|
| | MoT5 | MiLM | BM25 | NAIL |
| MS-Marco | 0.398 | **0.401** | 0.228 | 0.377 |
| Arguana | 0.288 | 0.415 | 0.472 | **0.522** |
| Climate-Fever | **0.28** | 0.24 | 0.186 | 0.206 |
| DBPedia-entity | 0.478 | **0.542** | 0.320 | 0.376 |
| Fever | **0.85** | 0.802 | 0.650 | 0.692 |
| FiQA-2018 | **0.514** | 0.334 | 0.254 | 0.411 |
| HotPotQA | **0.756** | 0.712 | 0.602 | 0.644 |
| NFCorpus | **0.384** | 0.36 | 0.343 | 0.367 |
| Natural Questions | **0.633** | 0.53 | 0.326 | 0.487 |
| SciDocs | **0.197** | 0.164 | 0.165 | 0.160 |
| SciFact | **0.777** | 0.682 | 0.691 | 0.710 |
| Trec-Covid | **0.795** | 0.722 | 0.688 | 0.766 |
| Touché 2020 | 0.3 | 0.349 | 0.347 | 0.240 |
| BEIR Avg | **0.511** | 0.481 | 0.405 | 0.458 |
| BEIR - MS-Marco | **0.521** | 0.488 | 0.420 | 0.465 |
| Total FLOPS | $10^{13}$ | $10^{12}$ | 0 | $10^4$ |

Table 2: BEIR results on reranking task (top 100 results from BM25). Note that we use the BM25 candidates from the ElasticSearch system. Results for all systems, Mo(no)T5-(3B), Mi(ni)LM(-L6), and BM25 are copied from the BEIR reranking leaderboard. Note MS-MARCO is in-domain for the trained models.

et al., 2020) and MiniLM-L6 [1]. MiniLM-L6 is a BERT-based models trained on MS-MARCO using a cross-encoder classifier. MonoT5-3B uses a T5-based model fine-tuned on MS-MARCO, using a generative loss for reranking.

**Results** Table 2 shows the reranking results. The baseline comparison for NAIL's performance here is BM25 alone: using BM25 without a reranker is the only other method that does not need to run a neural network for each query. We see that NAIL improves over BM25 fairly consistently. The improvement on MS-MARCO, which has in-domain training data, is especially striking. On BEIR, NAIL improves performance on 10 out of the 12 datasets increasing the average score by over 5 points.

While cross-encoder models are more powerful, they are also more expensive. Cross-encoder models must run inference on all 100 documents for each query. Thus, NAIL uses 8 to 9 orders of magnitude fewer FLOPS than cross encoder models, corresponding to almost 1 trillion fewer FLOPS for a single query. Moreover, NAIL significantly closes the gap between the BM25 baseline and the top performing cross-encoder rerankers, capturing 86% of the gains on MS MARCO and 45% of the gains on the broader suite of BEIR tasks. Thus, it presents an attractive alternative to expensive

---

[1] huggingface.co/cross-encoder/ms-marco-MiniLM-L-6-v2

rerankers when compute is limited.

## 5.2 Full Corpus Retrieval

In the *full corpus retrieval* task, each system must retrieve and rank from each dataset's corpus.

Because NAIL is very cheap to run as a reranker, it is reasonable to compare the BM25+NAIL results from Section 5.1 to direct retrieval systems that do not include a reranking step, but typically consume many orders of magnitude more FLOPs at query time. Table 3 presents this comparison.

As NAIL could be used to populate an inverted index, we investigate how well NAIL works when scoring all candidates in the corpus, which is an upper-bound for a NAIL-only retrieval system. These results are presented as NAIL-exh in Table 3.

We later present a brief investigation into the effect of sparsification of the NAIL output, to further understand the potential for using NAIL to populate a sparse inverted index for retrieval.

**Baselines**  For full retrieval, we compare NAIL to lexical-based and dual-encoder systems.

GTR-XXL (Ni et al., 2021) is one of the largest and best performing dual-encoder systems publicly available. It is pre-trained on a large, non-public, corpus of 2 billion QA pairs scraped from the web, and fine-tuned on MS-MARCO. Contriever is a dual-encoder system which employs novel self-supervised pretraining task (Izacard et al., 2021) and is fine-tuned on MS-MARCO; we describe it in more detail in Section 5.4.

SPLADE v2 (Formal et al., 2021a) develops query and document encoders to produce sparse representations, differing from dense dual-encoders systems. The query and document representations in SPLADE v2 are used for slightly different objectives. The query encoder is used to perform query expansion, and the document encoder is used to produce sparse representations for indexing. This system is trained via distillation of a cross-encoder reranker, and finally fine-tuned on MS-MARCO.

Colbert v2 adopts a late interaction model that produces multi-vector representations for both documents and passages. In this model, per-token affinity between query and document tokens are scored using per-token representations. It is trained via distillation of a cross-encoder reranker.

Besides BM25 and NAIL, SPLADE-doc$^+$ is the only other retriever that does not require neural network inference at query time. This model is a variant of SPLADE v2 where the query encoder is dropped, and only the document encoder is

used (Lassance and Clinchant, 2022). As with SPLADE v2, SPLADE-doc$^+$ is trained using distillation of cross-encoder reranker, with additional fine-tuning on MS-MARCO.

**Results**  Table 3 shows the results for nDCG@10 and recall@100 on BEIR full corpus retrieval for all systems that report it. We stratify the results into two sets, (1) MS-MARCO, which with the exception of BM25, is used as a training dataset, and (2) the average over all the other BEIR datasets, which are evaluated as zero-shot.

On the out-of-domain BEIR tasks, BM25+NAIL beats all but one of the neural retrieval systems, despite not encoding the query with a neural network and being limited in recall to BM25. Additionally, we note that NAIL-exh outperforms all other retrieval systems according to the recall@100 metric, suggesting potential for a NAIL-based retriever that uses NAIL to populate an inverted index. However, given the lower nDCG@10 than BM25+NAIL, this may only be worthwhile to implement if combined with a different reranker. Note that while recall@100 is highest for NAIL on the out-of-domain BEIR tasks, NAIL does worse than other models like GTR-XXL on the in-domain MSMARCO task. This is in part due to the training recipes used by other work to optimize for MS-MARCO performance, including model distillation and large non-public corpora of QA pairs.

SPLADE-doc also does not require a query-time encoder. We observe that NAIL lags on the in-domain evaluation but outperforms SPLADE-doc on both metrics of the zero-shot datasets in BEIR. As with many of the other retrievers, SPLADE-doc was distilled from a cross-attention reranker teacher, which may account for this in-domain gain in performance (Gao and Callan, 2022; Formal et al., 2022).

## 5.3 Comparison to Term Weighting Models

In this work we are primarily interested in the zero-shot multi-domain retrieval task represented by BEIR. However Table 4 also contains a comparison to other recent systems that use LMs to compute term weights, using the in-domain MS-MARCO passage retrieval task that they focused on. For NAIL, we report both the version which uses BM25 retrievals (in that case, the recall metric is derived from the BM25 system) and the system described in the previous section which uses exhaustive scoring. The results demonstrate that both NAIL-exh and BM25+NAIL outperform the other term weight-

| Metric | | Dual encoder | | Query encoder | | Lexical (no inf. net.) | | | |
|---|---|---|---|---|---|---|---|---|---|
| | | GTR-XXL | Contriever | SPLADE v2 | Colbert v2 | BM25 | SPLADE-doc$^+$ | NAIL-exh | BM25+NAIL |
| MS-MARCO | nDCG@10 | **0.442** | 0.407 | 0.433 | — | 0.228 | 0.431 | 0.396 | 0.377 |
| | recall@100 | **91.6** | 89.1 | — | — | 66.0 | 88.4 | 89.5 | 66.0 |
| Other BEIR | nDCG@10 | 0.459 | 0.445 | — | **0.469** | 0.420 | 0.429 | 0.432 | 0.465 |
| (avg. over 12 datasets) | recall@100 | 64.4 | 64.4 | — | — | 64.6 | 61.8 | **66.5** | 64.6 |
| Pt. w/ large QA corpus | | Yes | No | No | No | No | No | No | No |
| Pt. w/ distillation | | No | No | Yes | Yes | No | Yes | No | No |
| Pt. w/ self-supervision | | No | Yes | No | No | No | No | Yes | Yes |

Table 3: BEIR nDCG@10 and recall@100 results on the *full retrieval* task. The SPLADE-doc$^+$ results are previously unpublished, corresponding to the model described in (Lassance and Clinchant, 2022), and obtained via correspondence with authors. Other numbers are obtained from their respective publications.

| metric | DeepCT | DeepImpact* | COIL-tok | uniCOIL | SPLADE-doc | BM25+NAIL | NAIL-exh |
|---|---|---|---|---|---|---|---|
| MRR@10 | 0.243 | 0.326 | 0.341 | 0.315 | 0.322 | **0.363** | 0.356 |
| Recall@1000 | 0.913 | 0.948 | 0.949 | - | 0.946 | 0.814 | **0.981** |

Table 4: Evaluation on the MS-MARCO dev set for passage ranking task. Numbers reported are taken from corresponding publications: DeepCT (Dai and Callan, 2020), DeepImpact (Mallia et al., 2021), COIL-tok (Gao et al., 2021), uniCOIL (Lin and Ma, 2021), SPLADE-doc (Formal et al., 2021a). Results are obtained without document expansion, except for DeepImpact which includes terms from doc2query-T5 (Nogueira et al., 2019a).

ing models presented on the MRR@10 metric for the MS-MARCO passage ranking task. With respect to recall, NAIL-exh clearly improves over the previous systems. Exhaustive scoring is much more expensive than the other systems shown; however, given the sparsification results shown in Figure 3, we believe a sparse version of NAIL would be competitive with the models presented.

## 5.4 Comparison to Contriever

There are several confounding factors in comparing the systems presented in Tables 2 and 3. As mentioned, each system uses different training recipes and training data while also having slightly different architectures. Training techniques presented in the baselines presented in this work include unsupervised pretraining, hard negative mining, and distillation from a cross-attention teacher. These factors can make it difficult to pinpoint the cause of the variance in performance across models.

However, NAIL and Contriever (Izacard et al., 2021) share training recipes to a large extent, having both a similar pretraining stage followed by fine-tuning on MS-MARCO. Contriever is a recently introduced dual-encoder model that inspired the pretraining task in this work. However, architecturally, NAIL and Contriever are quite different. NAIL's query representation is not learned and is tied to the fixed set of vocabulary terms; this approach is potentially less powerful than a fully learned dense representation.

The summary of the comparison is available in

Table 8 (Appendix E). We observe that on the BEIR reranking task, NAIL matches both the in-domain and zero-shot performance of the Contriever model, despite lacking a query time neural network. Without using BM25 for initial retrievals, both methods perform slightly worse on nDCG@10 for the zero-shot BEIR tasks, but they remain comparable.

## 5.5 Performance versus query-time FLOPS

We have motivated this work by asking how much can we leverage large language models at indexing time while making query time computational costs small enough for a commodity CPU. As the results in this section show, there are tradeoffs between reranking improvements and computational costs. To illustrate this tradeoff, we present results of percentage nDGC@10 improvement over BM25 versus query-time FLOPS in Figure 4 (Appendix D). In general, we think lexicalized approaches like NAIL provide an interesting point on this curve, where much higher performance than BM25 can be achieved for only a small amount more compute. Note that Lassance and Clinchant (2022) discuss smaller versions of Splade; see Table 1 for the approximate reduction.

## 6 Concluding Remarks

We introduce a new model for sparse, lexicalized retrieval, called NAIL that adapts expensive pretrained sequence-to-sequence language models for document indexing. The main elements of NAIL are (1) the use of a non-autoregressive decoder, (2)

the use of vocabulary based representation for documents and queries, (3) a self-supervised training approach that is critical for good performance.

With NAIL, we focus on offloading all neural computation to indexing time, allowing serving to operate cheaply and without the use of accelerators. Evaluating retrieval on BEIR, we show that the NAIL approach is as effective as recent dual-encoder systems and captures up to $86\%$ of the performance gains of a cross-attention model on MS-MARCO while being able to serve requests on commodity CPUs.

## Limitations

The work presented in this paper contains several limitations. In this section we focus on limitations relating to (1) the choice of document representation (vocabulary-based vector of weights) and (2) empirical analysis using BEIR suite of datasets.

As described in Section 4, we inherit the vocabulary from the T5 models as basis for our document representation. This choice limits the applicability of NAIL in various ways:

1. The vocabulary is derived from an English-focused portion of the web. As a consequence, there are very few non-English word pieces encoded in the vocabulary, such as Unicode and other scripts. We expect this will have a significant, but unknown, impact on applying our system to non-English text.

2. In order to better support multi-lingual retrieval, we expect that the vocabulary of the model will need to be extended. For example, the multi-lingual T5, (mT5, Xue et al. 2021) contains 250 thousand items, an almost 8-fold increase compared to T5. This work does not study what the impact of vocabulary size increase can be on the quality of document representations and subsequently, on retrieval performance.

3. Unlike learned dense representations, our vocabulary-based representations may have more limited representational power. Recent work demonstrate that even in the case of learned dense representations, multiple representations can improve model performance (Lee et al., 2023; Zhou and Devlin, 2021). This work also does not evaluate the upper-bound on such vocabulary-based representations.

We believe the BEIR suite of datasets presents an improvement over prior text-based retrieval for QA, particularly focusing on a wider range of datasets and in zero-shot setting. Nonetheless, BEIR does not cover some domains in which NAIL may be under-perform. Beyond multi-linguality discussed above, we do not know how our model behaves when needing to reason about numbers or programming, or other domains of text which typically do not tokenize well.

This paper demonstrates that NAIL is competitive with other model expensive and complex neural retrieval systems. However, we do not present a highly optimized implementation of NAIL as a standalone retriever. An efficient implementation based on an inverted index is needed before NAIL can be used for end-to-end retrieval in high-traffic applications. Further work in sparsification of document representations (see Appendix C) is not explored in this work and is likely needed to achieve efficient indexing.

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

## A Qualitative Analysis

In this section, we present a qualitative analysis of the tokens that score highest according to the NAIL model for a given input. We choose the Natural Questions (NQ) subset of the BEIR benchmark for this analysis, as the queries tend to be complete questions that are easily interpretable. Table 5 shows the percentage of NAIL's top predicted tokens that appear in the passage input to the NAIL model along with the gold query that is paired with this passage in the NQ development set. Figure 2 presents the top predicted terms for a randomly sampled set of passages.

|  | top-100 | | top-1000 | |
|---|---|---|---|---|
|  | pretrained | tuned | pretrained | tuned |
| query | 53 | 74 | 85 | 94 |
| passage | 65 | 54 | 90 | 88 |

Table 5: Percent of NQ query and gold passage tokens contained in the top 100 and 1000 scores from NAIL.

Almost all of the tokens in both the input passages and the unseen query are present in NAIL's top 1000 predictions (Table 5). However, tuning towards MS-MARCO significantly increases the number of query tokens predicted in the top 100 and 1000 positions, while simultaneously reducing the number of passage tokens predicted. This is unsurprising: the fine-tuning stage represents a domain shift from the pre-training task, which is predicting document tokens, toward predicting query tokens. One indication of this shift is the increase in the prevalence of 'wh' words (what, who, where) in the top terms from the finetuned model in Figure 2.

Figure 2 also illustrates some other interesting shifts in NAIL's output during fine-tuning. For example, in Example (3) the pre-trained model predicts many dates associated with the Eagles (e.g., album release years). These are likely to occur in adjacent passages in the same document as the input passage, so they are good predictions for the pre-training task (Section 4.1). However, they are very unlikely to occur in queries associated with the input passage, and thus they are replaced in the fine-tuned predictions with terms that are more likely to occur in queries targeting the passage ('sang', 'sing', 'wrote', 'who', 'released').

Figure 2 also illustrates NAIL's ability to predict the type of query that is likely to be paired with a given passage. Passages containing definitions, such as the one presented in Example (1), are

highly associated with the wh-word 'what'. On the other hand, passages about individuals or groups of individuals (Examples (3) and (4)) are more highly associated with 'who'.

Finally, the predicted terms in Figure 2 contain a lot of small surface-form variations of the same root word, with different segmentations and capitalizations treated separately by the query tokenizer. For example, the tokens 'chic', 'chi', 'CHI', 'Ch', 'ch', 'CH' in Example (2) are all probably coming from different forms of the word 'Chicago' presented in different contexts. This redundancy illustrates a drawback of our featurization: unlike neural models, it does not abstract over diverse surface forms. Future work could examine more efficient and discriminative featurizations than the tokenization used in this work.

## B Alternate training recipes

Our primary goal has been to determine the extent to which the performance of an expensive neural network can be captured in a fast, sparse, featurization for general purpose retrieval. Subsequently, we have prioritized a training recipe that is aligned with previous work and well suited to the multi-domain BEIR task. However, the performance of learned retrievers as rerankers is very sensitive to the exact nature of the training recipe, and in this section we present analyses of the choices we made, and the associated trade-offs on BEIR and MSMARCO performance.

### B.1 Hard-negative selection for fine-tuning

One key choice in contrastive learning is the distribution of negative examples used in Equation 3. This is commonly a combination of hard negatives, which are chosen to be challenging for a single example, and batch negatives, which are drawn from the distribution of all positive and hard-negative candidates across training examples (Karpukhin et al., 2020; Xiong et al., 2020; Qu et al., 2021).

| # of hard negatives | MS-MARCO nDCG@10 | Avg. BEIR nDCG@10 |
|---|---|---|
| 3 | 0.377 | 0.465 |
| 7 | 0.378 | 0.461 |
| 15 | 0.391 | 0.460 |
| 31 | 0.394 | 0.457 |
| 63 | 0.397 | 0.457 |

Table 6: Effect of varying the number of hard negatives on reranking evaluation for MS-MARCO and BEIR. The BEIR average is computed without MS-MARCO.

| NQ query

(not shown to model) | NQ gold passage

(sole input to model) | Top terms from pre-trained model
(predictions) | Top terms from finetuned model
(predictions) |
|---|---|---|---|
| (1) what is non controlling interest on balance sheet | In accounting, minority interest (or non-controlling interest) is the portion of a subsidiary corporation's stock that is not owned by the parent corporation. The magnitude of the minority interest in the subsidiary company is generally less than 50% of outstanding shares, or the corporation would generally cease to be a subsidiary of the parent.[1] | minority, controlling, passage, subsidiary, Minor, 50%, interest, accounting, control, 1, Interest, question, Accounting, control, non, Non, generally, 1., owned, Answer, subsidiaries | minority, interest, Definition, ities, controlling, Non, non, what, ity, interests, control, own, Interest, Minor, shares, owned, ownership, Both, accounting, interest, stock, does, mean, Control |
| (2) how many episodes are in chicago fire season 4 | The fourth season of Chicago Fire, an American drama television series with executive producer Dick Wolf, and producers Derek Haas, Michael Brandt, and Matt Olmstead, was ordered on February 5, 2015, by NBC,[1] and premiered on October 13, 2015 and concluded on May 17, 2016.[2] The season contained 23 episodes.[3] | NBC, stead, Wolf, Fire, fourth, Chicago, season, 2016, firefighters, concluded, episodes, Ha, 4, 3, 5, contained, aire, 6, 23, 4., 4,, premiere, characters, episode, Identify, fire | chic, fire, fourth, seasons, chi, season, Fire, shows, 4, ich, air, CHI, ch, Ch, four, episodes, when, series, Season, CH, show, Chicago, Fi, 4, chie, firefighters, NBC, four, aire |
| (3) who sings love will keep us alive by the eagle | "Love Will Keep Us Alive" is a song written by Jim Capaldi, Paul Carrack, and Peter Vale, and produced by the Eagles, Elliot Scheiner, and Rob Jacobs. It was first performed by the Eagles in 1994, during their Hell Freezes Over reunion tour, with lead vocals by bassist Timothy B. Schmit. | Eagle, Schm, live, 1994, Love, Keep, Us, alive, song, Free, performed, lyrics, Glen, 1976, 1995, rack, IVE, 1977, during, 1975, 1993, keep, 1972, 1974, 1996, 1997, Don, album | Eagle, alive, sang, love, live, song, Cap, written, keep, sing, live, wrote, lov, Love, kept, Will, who, will, hell, keep, ll, keeps, Live, tim, Us, gle, singer, songs, cap, IVE, Car, written |
| (4) nitty gritty dirt band fishin in the dark album | "Fishin' in the Dark" is a song written by Wendy Waldman and Jim Photoglo and recorded by American country music group The Nitty Gritty Dirt Band. It was released in June 1987 as the second single from their album Hold On.[1] It reached number-one on the U.S. and Canadian country charts. It was the band's third number-one single on the U.S. country music charts and the second in Canada. After it became available for download, it has sold over a million digital copies by 2015.[2] It was certified Platinum by the RIAA on September 12, 2014.[3] | Wald, itty, glo, hin, Dir, Dark, Wendy, Fi, RIA, fishing, dark, 1987, song, 5, 4, million, 3, Gr, Fish, 5., single, became, Hold, Band, number, 1986, 1, (4), 6, country, band, reached, Jim, 500,000, 1988 | hin, fi, dark, itty, fishing, song, Dir, Wald, sang, sing, hold, wald, fish, ?, ity, Fish, band, gg, who, shing, band, hit, dir, songs, held, ies, Wendy, singer, dirty, Hold, released, Band, ISH, dirt, country, fish, Dark, Song, ities, written, music, single, Country, ddy, when, wrote |

Figure 2: Sample of top token predictions from pre-trained only and pre-trained+fine-tuned NAIL models. The table shows a few evaluation examplars from the Natural Questions evaluation set included in BEIR. We display the corresponding question associated with the answer passage for the benefit of the reader, but this is not shown to the model. We have explicitly removed stop words and non-words (control sequences). Note that due to the the use of SentencePiece tokenizer (Kudo and Richardson, 2018), tokens do not necessarily correspond to full words.

| nDCG@10 | Finetuned only | Pretrained only only | Pretrained + Finetuned |
|---------|----------------|----------------------|------------------------|
| MSMARCO | 0.367 | 0.212 | 0.377 |
| BEIR | 0.422 | 0.416 | 0.465 |

Table 7: Effect of pretraining on NAIL for the BEIR reranking task. The BEIR nDCG@10 metric corresponds to average score of datasets excluding MS-MARCO.

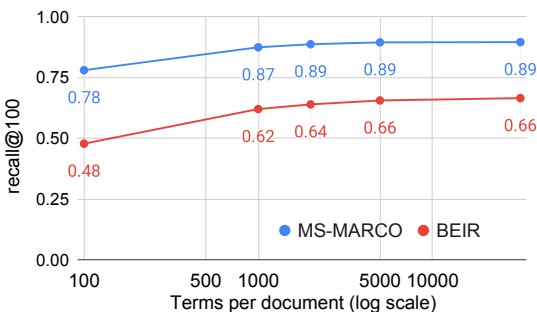

Figure 3: Effect of sparsification of document representation on recall@100, using a top-k strategy.

Our pretraining task (described in Section 4.1) does not use hard negatives; however, the MS-MARCO fine-tuning task includes hard negatives created by running BM25 retrieval over the set of candidate passages. Table 6 shows how BEIR and MS-MARCO results change as we change the number of MS-MARCO hard-negatives that we sample during fine tuning. As this number increases, the MS-MARCO performance also increases until it matches the performance of the cross-attention rerankers in Table 2 when 63 hard negatives are sampled for each training example. However, increasing the number of MS-MARCO hard negatives also hurts BEIR performance.

## B.2 Effects of pretraining and fine-tuning

The training recipe, presented in Section 4.1, has two stages beyond the language model training from Raffel et al. (2020). Table 7 shows that both stages benefit both the BEIR and MSMARCO results. However, NAIL still yields a nice improvement over BM25 across the BEIR tasks using only the pre-training task. This is encouraging because these data are heuristically generated rather than relying on human relevance labels, so they can be trivially applied to new domains. The MS-MARCO results are unsurprisingly more dependent on fine-tuning on MS-MARCO. Pre-trained NAIL does not outperform BM25 on MS-MARCO without fine-tuning. More sophisticated methods of synthetic training data generation, such as Promptagator(Dai et al., 2022), could also help improve NAIL further, but we leave this to future work.

## C Sparsification

To further explore the potential for using NAIL for full retrieval, we experiment with a naive approach to sparsifying NAIL document representations. Specifically, we simply order tokens by their scores and keep the *top-k* scoring tokens.

Figure 3 demonstrates the effect on the recall@100 metric of reducing the number of terms per document from the original vocabulary of 32

thousand tokens down to 100 tokens. For both MS-MARCO and other BEIR datasets, recall@100 falls considerably when using only the top 100 tokens. Nonetheless, with only two thousand tokens we are able to maintain the same level of performance for MS-MARCO and roughly 97% of the recall performance on BEIR. This observation, along with the results in Table 3, suggest that NAIL could be used to populate an efficient inverted index for retrieval, with little loss of recall. Such an index could serve as a more powerful alternative to BM25. We leave this to future work.

## D Performance versus query-time FLOPS

Figure 4 illustrates different systems with varying tradeoff between computational cost and retrieval performance. See Section 5.5 for the discussion on this figure.

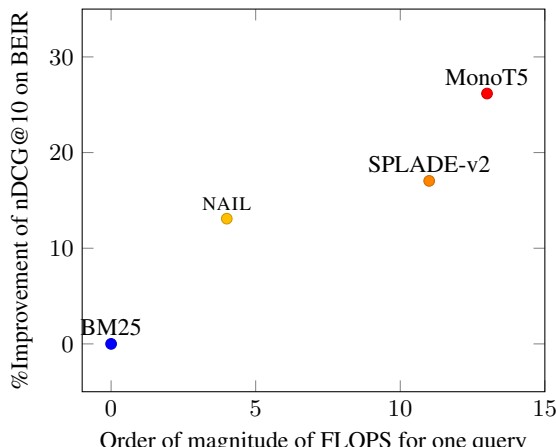

Figure 4: Improvement over BM25 and extra FLOPS to score one query on the BEIR retrieval task. The NAIL and MonoT5 use BM25 retrievals; SPLADE-v2 uses its own retrievals over the full corpus. Note that the vast majority of the computation for SPLADE and dual encoders is in encoding the query; reranking BM25 retrievals would not reduce computation.

## E   Comparison to Contriever

Table 8 compares the reranking performance of the Contriever system with NAIL. See Section 5.4 for the discussion on this comparison.

| nDCG@10 | Contriever | BM25+Contr. | NAIL-exh | BM25+NAIL |
|---|---|---|---|---|
| MS-MARCO | 0.407 | 0.371 | 0.396 | 0.377 |
| Avg. BEIR | 0.445 | 0.463 | 0.432 | 0.465 |

Table 8: Comparison of Contriever and NAIL on BEIR and MS-MARCO. We obtain Contriever reranking performance by using their released model and ranking the same set of BM25 candidates as NAIL. The average BEIR nDCG@10 does not include MS-MARCO.