# OpenReview forum: "NAIL: Lexical Retrieval Indices with Efficient Non-Autoregressive Decoders"
_EMNLP/2023/Conference — EMNLP 2023 Main_

### Official Review · Reviewer_w6rA · 2023-08-11

**Soundness:** 4

**Excitement:**

4: Strong: This paper deepens the understanding of some phenomenon or lowers the barriers to an existing research direction.

**Paper Topic And Main Contributions:**

The paper introduces a novel retrieval system named NAIL to accelerate the inference time by pushing workload to indexing time. The idea is to precompute the document indexing using a non-autoregressive decoder to score tokens in parallel and only perform tokenization of queries at the retrieval time. Experiments on the BEIR benchmark and MS-MARCO show that NAIL can significantly maintain performance compared to state-of-the-art cross-encoder and dual-encoder systems while being more computationally efficient. It pushes further the progress of sparse retrieval and is of great practical value in modern retrieval systems.

**Questions For The Authors:**

A. As shown in Figure 1, the input to the Encoder is prompt+passage. Does the prompt design indeed affect the performance?
B. The Authors used multi-hot vectors for query representation, seemingly ignoring the word order information that contributes to semantics. Can you elaborate more on this concern?
C. Should the defined contrastive loss in Equation 3 be maximized instead of minimized?

**Reasons To Accept:**

The research question is well-motivated and properly justified. Using a non-autoregressive decoder for document indexing is novel and non-trivial. Thorough experiments and analysis confirm both the effectiveness and efficiency of the proposed retrieval system. The paper is clear-written and easy to follow.

**Reasons To Reject:**

N/A

**Reproducibility:**

4: Could mostly reproduce the results, but there may be some variation because of sample variance or minor variations in their interpretation of the protocol or method.

**Reviewer Confidence:**

3: Pretty sure, but there's a chance I missed something. Although I have a good feel for this area in general, I did not carefully check the paper's details, e.g., the math, experimental design, or novelty.

**Typos Grammar Style And Presentation Improvements:**

- Equation 2:  unmatched parenthesis

---

> ### Author Rebuttal · Authors · 2023-08-29
>
> We thank reviewer w6rA for their positive and encouraging comments. We are happy to hear that the reviewer enjoyed the experimental work presented and that the paper was easy to follow. We agree that the use of non-autoregressive decoder for indexing is novel and worthy of more research. We hope the community can benefit from this initial work which we focused primarily on query-time efficiency.
>
> Below we address the three questions the reviewer has posed:
>
>
> **Question A (prompt design)**: Unfortunately, we have not experimented extensively with prompt design, so we cannot provide good experimental evidence on this matter. Nonetheless, it is our belief that the specific prompt used is much less influential on model behavior when the model is trained on a given task, versus on zero- and few-shot settings where it has been shown that the prompt design is quite important.
>
> Specifically, in our case, we have added a prompt to the input so that the model can differentiate between tasks: the two pre-training tasks (inverse cloze and independent cropping) and the fine-tuning task (query term prediction). Typically for task differentiation, when the model is trained on many examples, the prompt can be as simple as a marker or simple string to allow the model to differentiate between the tasks.
>
>
> **Question B (word order)**: We briefly touch upon the issue of order agnostic representation in the “Limitations” section of the paper. Given our experience with the model, and specifically looking at many error cases in model ranking, we believe this is currently the main weakness in the proposed lexicalized (token-based) query representation. There are many cases where n-grams, whether denoting an entity name, idiomatic expression, complex predicate, etc., cannot be easily inferred from an unordered bag of tokens (for example: “house”, “door”, “white”, “blue”, “with”).  Nonetheless, we believe that studying an “extreme” case of single token representations without much processing of the query (beyond tokenization) is a compelling design point to study. We hope it inspires others to go beyond our simple query representation to explore other low-cost options.
>
> For future work, we think it is likely the most interesting way to extend this line of work is to design an efficient representation that can go beyond single tokens. Precisely because it can enable more representational power than unigram representations. A few ideas include:
>
> It would be interesting to study the gap between representational power of 1-gram vs. n-grams query representations for retrieval and/or ranking. This can be done with a simulated oracle or some other clever experimental design.
> We also think it could be useful to study efficient ways to match precomputed n-gram representations. One option in this line of work could be to extend the base model vocabulary from 10s of thousands of sentence-pieces to millions of sentence-pieces (e.g., Liang et al 2023 - https://arxiv.org/abs/2301.10472).
> Study the use of denoising non-autoregressive decoders, where a large number (e.g., 100-1000) of n-grams can be sampled in an efficient manner and used for indexing instead of along with unigram representations.
>
> **Question 3 (Equation 3 max/min)**: Thank you for catching this error, we agree with your comment.  Equation 3 defines a contrastive loss based on the score function described in Equation 1. As described, scores with higher values indicate better compatibility between query and document. As such, minimizing Equation 3 is incorrect, we agree with the reviewer. We will fix Equation 3 to invert the sign, making it closer to the code that trains the model, specifically:
>
> $$
> \mathcal{L} = - \langle \phi_q(q_i), \phi_d(p^+) \rangle +  {\rm log} \sum_{p' \in \mathbf{p}_i} {\rm exp}(\langle \phi_q(q_i), \phi_d(p')\rangle)
> $$
>
> So that minimizing the sign inverted version of Equation 3 is the correct training procedure.

---

### Official Review · Reviewer_Esnj · 2023-08-11

**Soundness:** 4

**Excitement:**

4: Strong: This paper deepens the understanding of some phenomenon or lowers the barriers to an existing research direction.

**Paper Topic And Main Contributions:**

This work introduces a new model called NAIL for sparse, lexicalized retrieval. It adapts a non-autoregressive decoder for document indexing, and predicting scores on vocabulary instead of generating token sequences, which largely reduces the computation cost. Experiments show that NAIL largely reduces the computation cost while capturing up to 86% performance gains of cross-attention model.

**Reasons To Accept:**

1. The evaluation results really demonstrate the good performance of the proposed method.
2. The idea of predicting on vocabulary instead of generating tokens is interesting.

**Reasons To Reject:**

N/A

**Reproducibility:**

4: Could mostly reproduce the results, but there may be some variation because of sample variance or minor variations in their interpretation of the protocol or method.

**Reviewer Confidence:**

4: Quite sure. I tried to check the important points carefully. It's unlikely, though conceivable, that I missed something that should affect my ratings.

---

> ### Author Rebuttal · Authors · 2023-08-29
>
> We thank reviewer Esnj for their positive review. We agree that this work has surprising results in that predicting vocabulary items (in lieu of generation) can yield such positive results, even when compared to more powerful techniques. The performance of the studied technique is shown to be robust across a range of retrieval tasks, strengthening the curiosity of this finding.

---

### Official Review · Reviewer_oAAC · 2023-08-12

**Soundness:** 3

**Excitement:**

3: Ambivalent: It has merits (e.g., it reports state-of-the-art results, the idea is nice), but there are key weaknesses (e.g., it describes incremental work), and it can significantly benefit from another round of revision. However, I won't object to accepting it if my co-reviewers champion it.

**Paper Topic And Main Contributions:**

This paper provides a sparse retrieval framework to leverage LLM models to learn a representation for documents at the indexing time with the goal of omitting the high computation that LLM-based retrieval models require at the inference time. Authors train a seq-2-seq model with a modification to the decoder module to generate tokens in a non-auto-regressive approach to represent documents. The model outputs a probability distribution over all vocabulary at each position and the final distribution over the vocabulary is calculated by taking the maximum over all positions. The authors train the model in two steps: 1) Pre-training on inverse cloze and in-dependent cropping task and 2) fine-tuning on the MsMarco passage retrieval task. The authors conduct experiments on MSMarco as well as the BEIR benchmark which consists of several retrieval datasets in different domains. They compare their model with the cross-encoder and dual-encoder approaches that were previously investigated on these datasets. The experiments show NAIL achieves up to 86% of the performance of other transformer-based retrieval approaches (including cross-encoder, dual encoder, and spare retrieval) by reducing the FLOPs from 10^13 in a cross-encoder architecture to 10^4.

**Reasons To Accept:**

The paper investigates the important problem of leveraging LLM efficiently in retrieval in the case in which computationally intensive approaches are not feasible at the time of inference. The authors conduct a comprehensive literature review. The authors compare their proposed models with different approaches in neural retrieval i.e. cross-encoder, dual encoder, and sparse retrieval.  The authors discussed the limitation of their proposed model from different aspects. The model's effectiveness is reasonable considering the reduction in FLOPs relative to other neural retrieval methods.

**Reasons To Reject:**

The main contribution of the paper is reducing the computational cost at the inference which is measured by FLOPs. However, the authors did not provide any experimentations detail of how they measure FLOPs which makes the reproducibility and confirmation of results challenging.

The final step of similarity between query representation and document representation is not clear as well as how an index was created using the NAIL-Exhaust has not been explained. Also, the intuition behind why representing documents based on the non-autoregressive token prediction and comparing to the query token representation is not clearly explained.

The writing and structure of the paper need to be improved: there are instances of undefined parameters and confusing equations (32k^16 = 10^72, Section 3.1) in the paper as well as inconsistency in notations (passages once are referred to as d and later referred to as p in section 3.2)

The authors need to provide more rationale and discussion on why pre-training on two inverse cloze and in-dependent cropping task is needed and helpful for Nail.

Lastly, since the authors propose Nail as a mechanism for reducing computational cost at inference, comparing Nail with approaches using Distilled models can be good comparison to different approaches for reducing inference time computational cost complexity.

**Reproducibility:**

4: Could mostly reproduce the results, but there may be some variation because of sample variance or minor variations in their interpretation of the protocol or method.

**Reviewer Confidence:**

4: Quite sure. I tried to check the important points carefully. It's unlikely, though conceivable, that I missed something that should affect my ratings.

---

> ### Author Rebuttal · Authors · 2023-08-29
>
> We thank reviewer oAAC for a thorough review of our work. We are encouraged by the reviewer’s agreement that leveraging LLMs for retrieval, in an efficient manner, is an important problem. Nonetheless, the review provides low scores in excitement and soundness, listing a few reasons for rejection. Below, we touch upon each of the reasons mentioned, and hope that through these clarifications, the reviewer can reassess their evaluation of the work as we think most of the issues raised can be addressed with modest edits to the paper.
>
>
> **Flops computation**: We apologize for lack of clarity with respect to inference-time FLOPS computation we presented in the paper. There are two classes of models for which we present inference-time FLOPS: methods with neural inference and methods with lexical-based scores. For the neural-based models, as noted in the caption of Table 1, we adopted the methodology used in Clark et al 2020 (https://openreview.net/pdf?id=r1xMH1BtvB). Specifically, we used their open-source code at https://github.com/google-research/electra/blob/master/flops_computation.py. Their method covers all Transformers-based models we compare to in the paper.
>
> For lexical-based models, we compute the cost of scoring query terms, which typically use a constant and small (typically one or two) number of floating point operations per query term (word or token).
>
> We can make these computations more clear in the final version of the paper.
>
>
> **NAIL-Exhaust**: In Section 5.3 and Table 4 we present a proof of concept experiment to demonstrate the potential for using NAIL as an indexing system rather than using it as a reranker. As we describe in that section, we have not properly implemented an indexing scheme for retrieval, but instead use an *exhaustive* approach to retrieval where for each query, we score all documents in the corpus for that query. As we mention in the section, this exhaustive implementation is impractical. But we hope the results of the experiment are insightful and motivating for future designs of practical and sparse NAIL-based indexes.
>
> **32k^16 = 10^72, Section 3.1**: We can clarify this computation in the final version of the paper. But essentially, the first term comes from the size of the model’s English-based vocabulary, which has 32,768 items (32k). We must explicitly score each of these vocabulary items for each query term (which we limit query length to 16 tokens), so 32,768^16 (32k^16). Which is approximately 10^72 decodes.
>
> **Notation inconsistency**: Thank you for catching inconsistency in Section 3.2, specifically around equations 2 and 3. We will fix this inconsistency.
>
> **Two pre-training tasks**: We have not done an extensive analysis of the mix of pre-training tasks for NAIL. The main reason for this is that pre-training incurs a non-negligible cost which we seeked to minimize. Informally, we compared two pre-training setups: (a) the same setup as described in the Izacard et al 2022 (https://arxiv.org/abs/2112.09118) which applies only independent cropping task, and (b) created a mixture of the two tasks described in the paper (independent cropping and inverse cloze). We noticed that the combination of tasks performed slightly better in the query generation task. We stipulate that, unlike in the analysis done by Izacard et al., the query generation task is different from constructing dense document representations, and the inverse cloze task is more similar to query generation than the independent cropping task.
>
> **Distilled models**: We agree that distilled models are an alternative and interesting way to reduce inference-time costs of language models. Due to space constraints, we limited the number of comparisons we have presented in the paper, but we are open to expanding these comparisons further.
>
> In Table 1, we do present FLOPS information on a few distilled models for retrieval from the literature, including MiniLM-L6 (derivative of Wang et al 2020 - https://arxiv.org/abs/2002.10957), TinyBERT-L6 (derivative of Chen et al 2020 - https://arxiv.org/abs/2009.07531), and BERT-tiny (derivate of Turc at al 2019 - https://openreview.net/pdf?id=BJg7x1HFvB ).
>
> Out of these, we selected MiniLM-L6 as representative of these models since it performs the best according to the BEIR leaderboard at https://docs.google.com/spreadsheets/d/1L8aACyPaXrL8iEelJLGqlMqXKPX2oSP_R10pZoy77Ns/edit#gid=867044147. Results from this model are presented alongside our main results in Table 2 (MiLM column). Fundamentally, even the smallest of the distilled cross-encoder models (for example, BERT-tiny) is still orders of magnitude more expensive for inference than lexicalized approaches, and as Table 2 shows, provide only a couple of points improvement over our proposed NAIL reranker.

---

### Meta-Review · Area_Chair_9zMw · 2023-09-07

**Recommendation:** 4

**Metareview:**

All authors found merits in the submission, especially in the improvement of efficiency in reranking process. Although some reviewers pointed out the problem of failing to show FLOPs improvement explicitly, the authors described these information in the rebuttal. I strongly suggest the authors revise the manuscript per the comments of the reviewers.

---

### Decision · Program_Chairs · 2023-10-07

**Decision:**

Accept-Main

**Comment:**

All authors found merits in the submission, especially in the improvement of efficiency in reranking process. Although some reviewers pointed out the problem of failing to show FLOPs improvement explicitly, the authors described these information in the rebuttal. I strongly suggest the authors revise the manuscript per the comments of the reviewers.